# Intratumoral *SPP1*^+^*BCL2A1*^+^ Tumor-Associated Macrophages Predict Poor Response to PD1 Blockade

**DOI:** 10.3390/diagnostics15212680

**Published:** 2025-10-23

**Authors:** Chun-Hao Lai, Yu-Ping Hung, Po-Chun Tseng, Rahmat Dani Satria, Chiou-Feng Lin

**Affiliations:** 1Department of Microbiology and Immunology, School of Medicine, College of Medicine, Taipei Medical University, Taipei 110, Taiwan; 2Graduate Institute of Medical Sciences, College of Medicine, Taipei Medical University, Taipei 110, Taiwan; 3Core Laboratory of Immune Monitoring, Office of Research & Development, Taipei Medical University, Taipei 110, Taiwan; 4Department of Clinical Pathology and Laboratory Medicine, Faculty of Medicine, Public Health and Nursing, Universitas Gadjah Mada, Yogyakarta 55281, Indonesia; 5Clinical Laboratory Installation, Dr. Sardjito Central General Hospital, Yogyakarta 55281, Indonesia

**Keywords:** hepatocellular carcinoma, PD1, secreted phosphoprotein 1, Bcl-2-related protein A1, tumor-associated macrophages

## Abstract

**Background/Objectives:** Immune checkpoint blockade (ICB) has emerged as a promising therapeutic option for hepatocellular carcinoma (HCC), yet reliable biomarkers to predict clinical outcomes remain limited. Tumor-associated macrophages (TAMs) are increasingly recognized as key regulators of the tumor immune microenvironment. **Methods:** We interrogated a publicly available HCC single-cell RNA sequencing (scRNA-seq) dataset to characterize intratumoral immune cell subpopulations. Through unsupervised clustering and gene signature analysis, we identified a distinct subset of *SPP1* (secreted phosphoprotein 1, also known as osteopontin) and BCL2A1 (Bcl-2-related protein A1) double-positive TAMs. Their abundance was quantified and associated with patient outcomes. Further independent HCC transcriptomic datasets with annotated PD1-based ICB response status were used for examination. **Results:** Across the discovery (GSE149614; *n* = 10) cohort, elevated expression of intratumoral *SPP1*^+^*BCL2A1*^+^ TAMs was identified in HCC. In the ICB datasets (GSE151530; *n* = 4), patients with high *SPP1*^+^*BCL2A1*^+^ TAM expression further exhibited significantly poorer responses to ICB therapy. Further, the validation cohort (GSE206325; *n* = 18) confirmed these findings accordingly. Notably, these TAMs were expressed thoroughly within the immunosuppressive T-cell microenvironment in non-responders but were distinctly expressed among the cytotoxic T-cell responses in responders. **Conclusions:** Our findings identify *SPP1*^+^*BCL2A1*^+^ TAMs as a poor prognostic biomarker in HCC patients undergoing ICB therapy. By promoting an immunosuppressive microenvironment, *SPP1^+^BCL2A1^+^* TAMs, which are survival-advantaged, may represent both a predictive marker and a potential therapeutic target to enhance the efficacy of immunotherapy.

## 1. Introduction

Hepatocellular carcinoma (HCC) is the most common primary liver cancer and represents a leading cause of cancer-related mortality worldwide. The rising incidence of HCC is closely linked to chronic viral hepatitis, alcohol consumption, and metabolic disorders such as non-alcoholic steatohepatitis [1]. Despite advances in surgical resection, locoregional therapies, and systemic treatments, long-term survival outcomes remain poor, particularly for patients with advanced or unresectable disease [2]. Consequently, there has been an increasing focus on harnessing the immune system for therapeutic benefit, with immune checkpoint blockade (ICB) emerging as a promising approach [2,3]. In HCC, clinical trials have demonstrated durable responses to programmed death 1 (PD-1)/PD-Ligand 1 (PD-L1) blockade in a subset of patients. Nevertheless, the majority of HCC patients fail to achieve meaningful benefit [4], underscoring the need for reliable biomarkers that can predict responsiveness to immunotherapy and guide patient selection. Current biomarker strategies, such as PD-L1 expression, tumor mutational burden, and microsatellite instability, have shown limited predictive value in HCC [5]. Thus, elucidating the immune landscape of HCC and identifying tumor-intrinsic and microenvironmental determinants of therapeutic resistance remain pressing priorities.

Tumor-associated macrophages (TAMs) constitute a dominant component of the HCC tumor microenvironment. These cells exhibit remarkable phenotypic plasticity, ranging from pro-inflammatory and antitumorigenic states to immunosuppressive and tumor-promoting phenotypes. Increasing evidence suggests that TAMs play a role in promoting immune evasion, angiogenesis, and metastasis [6]. In particular, subsets of TAMs expressing immunosuppressive molecules, such as C1Q, CD163, or MARCO, have been linked to a poor prognosis in HCC and other cancers [7,8]. Yet, the functional heterogeneity of TAMs in relation to ICB response remains incompletely defined. In HCC, high intratumoral CD38^+^CD68^+^ TAMs are linked to more prolonged survival through enhanced IFN-γ production [9], while about 25% of tumors form an “immune-specific class” with high PD-L1 and cytolytic markers, subdivided into active (T cell/IFN-enriched) and exhausted (T-cell exhaustion, TGF-β, M2 macrophages) subtypes [10].

Secreted phosphoprotein 1 (SPP1), also known as osteopontin, is a multifunctional cytokine and extracellular matrix protein with established roles in inflammation, tissue remodeling, and cancer progression [11]. Elevated levels of circulating and tumor-derived SPP1 have been correlated with aggressive clinical features in HCC, including vascular invasion and metastasis [12]. Within the immune system, SPP1 is highly expressed by activated macrophages, where it contributes to shaping the tumor immune microenvironment as a tumour-immune barrier structure by promoting T-cell exhaustion and suppressing cytotoxic activity [13,14,15,16]. Analyses of single-cell RNA sequencing (scRNA-seq) datasets from HCC have revealed a distinct subset of TAMs characterized by high expression of SPP1. These *SPP1*^+^ TAMs are enriched in tumor tissues compared to normal liver, and their abundance has been associated with poor overall survival [15,17,18,19,20,21]. Notably, *SPP1*^+^ TAMs are specifically associated with resistance to ICB therapy in HCC [16]. Regarding the presence of *SPP1*^+^ TAM program inducing resistance to PD-1 blockade, clarifying the phenotype of such cells and identifying their regulation could provide critical insights into mechanisms of immunotherapy resistance and identify novel therapeutic targets. In this study, we interrogated multiple scRNA-seq transcriptomic datasets of HCC to define the presence and clinical significance of the newly identified subtype of intratumoral TAMs.

## 2. Materials and Methods

### 2.1. Single-Cell RNA Sequencing Dataset Analysis

We analyzed a publicly available hepatocellular carcinoma (HCC) single-cell RNA sequencing (scRNA-seq) dataset (GSE149614; *n* = 10, GSE151530; *n* = 4, and GSE206325; *n* = 18) [21,22,23], which included paired tumor and adjacent normal tissues without or with ICB therapy for discovery and validation. Raw expression matrices and cell annotation files were downloaded and preprocessed using Scanpy (v1.9.6) in a macOS Python 3.9 (Anaconda virtual environment). In validation cohort GSE206325, patient metadata (age, sex, and treatment) were available for all cases. All received cemiplimab. Responders (*n* = 6) were all male, ages 47–82; non-responders (*n* = 12) were mostly male, ages 45–77.

Quality control (QC) criteria included filtering cells with ≥200 detected genes, retaining genes expressed in ≥3 cells, and excluding cells with more than 5% mitochondrial content. The data were normalized to 10,000 counts per cell and then log-transformed. Highly variable genes (*n* = 2000) were selected using the Seurat vst method, followed by principal component analysis (PCA, 50 components). A nearest-neighbor graph (k = 10) was constructed based on the top 20 PCs, and Leiden clustering (resolution = 0.5) was performed. Uniform Manifold Approximation and Projection (UMAP) was applied for visualization.

Cluster-specific marker genes were identified using Wilcoxon rank-sum tests (Scanpy sc.tl.rank_genes_groups), with significant markers defined as false discovery rate (FDR) <0.05 and log-fold change >0.5. The top 20 upregulated genes per cluster were used for annotation. Cell-type identities were determined by cross-referencing canonical lineage markers with literature integration, aided by ChatGPT (OpenAI GPT-5) for marker interpretation (Appendix A) [24].

Differential gene expression analysis was performed with selective clusters as the target macrophage population. Volcano plots displayed significantly enriched genes, and dot plots illustrated canonical gene signatures. The abundance of secreted phosphoprotein 1 (SPP1^+^, osteopontin) and Bcl-2A1 (BCL2A1^+^) TAMs were quantified across tumor and normal specimens, and UMAP mapped their spatial distribution.

### 2.2. Association with ICB Therapy Response

To examine the clinical relevance of *SPP1*^+^*BCL2A1*^+^ TAMs, we analyzed an HCC transcriptomic dataset with annotated response status to PD-1-based ICB (GSE151530 and GSE206325). Patients were stratified into two groups: non-responders (NRs) and responders (Rs), based on clinical outcomes. Gene signature scoring (scanpy.tl.score_genes) was applied to quantify *SPP1*^+^*BCL2A1*^+^ TAMs enrichment, and scores were compared between groups. Statistical analysis assessed the associations between *SPP1*^+^*BCL2A1*+ TAM levels, treatment response, and prognosis, revealing a higher abundance of intratumoral *SPP1*^+^*BCL2A1*+ TAMs in non-responders.

### 2.3. Post-Treatment Immune Transcriptomic Profiling

To investigate post-ICB immune remodeling, we analyzed an independent dataset of HCC patients treated with PD-1 blockade (GSE206325). UMAP embedding and clustering were performed as described above. Comparisons between NR (*n* = 12) and R (*n* = 6) groups focused on immune cell composition, distribution of *SPP1*^+^*BCL2A1*^+^ TAMs, and their association with T-cell states. Quantification of cell-type proportions (normal vs. tumor, pre- vs. post-treatment, and NR vs. R) was visualized using horizontal bar plots. Subgroup analyses were conducted to evaluate the abundance of *SPP1*^+^*BCL2A1*^+^ TAMs relative to the phenotypes of immune cell subsets. Association analysis compared cell-type counts between Rs and NRs.

### 2.4. KEGG Pathway Enrichment Analysis for Longevity Signatures

To explore functional programs associated with treatment response, we conducted Kyoto Encyclopedia of Genes and Genomes (KEGG) pathway analysis using the gseapy (v1.1.2) and sc.tl.score_genes functions in Scanpy. Differentially expressed genes (DEGs) between non-responders (NRs; *n* = 12) and responders (Rs; *n* = 6) to anti–PD-1 therapy (GSE206325) were identified with Wilcoxon rank-sum tests (log_2_FC > 0.25). Enriched KEGG pathways were identified using overrepresentation analysis (ORA) and gene set enrichment analysis (GSEA), with significance defined as an adjusted *p*-value < 0.05.

Particular focus was placed on the longevity-regulating pathway, characterized by the enrichment of several key genes (FOXO3, SIRT1, MTOR, etc.), in which enrichment scores were calculated at the single-cell level and visualized as pathway activity scores. Comparisons were made between NR and R groups to evaluate whether upregulation of longevity-associated programs was linked to the persistence of immunosuppressive subsets or altered T-cell functionality. Longevity pathway scores were further calculated and compared between groups.

### 2.5. Statistical Analysis

All data analyses were performed in Python 3.9 within Jupyter Notebook (v7.4.4). Key packages included Scanpy (v1.9.6) for preprocessing, normalization, clustering, UMAP visualization, and gene set scoring; NumPy (v1.24.3) and SciPy (v1.10.1) for numerical and statistical analysis; Pandas (v2.0.3) for tabular processing; and Matplotlib (v3.7.1) and Seaborn (v0.12.2) for visualization (UMAP, volcano plots, dot plots, bar plots, and boxplots). Matplotlib’s path effects were used for text readability enhancement, and the Statannotations/Statannot packages were utilized to perform Mann–Whitney U tests and to annotate statistical significance in the figures. *p*-values < 0.05 were considered statistically significant, with annotations (* *p* < 0.05; ** *p* < 0.01; *** *p* < 0.001; *** *p* < 0.0001). A bootstrap-based power analysis was conducted to estimate the ability to detect group differences (e.g., responders vs. non-responders). For each group, cells were randomly resampled with replacement to match the original sample size. Mean expression levels, scores, or cell proportions were compared between groups, and statistical tests were applied in each iteration. This process was repeated 1000 times, and the proportion of iterations with *p* < 0.05 represented the estimated power.

## 3. Results

### 3.1. Single-Cell Identification of a New Subtype of SPP1^+^BCL2A1^+^ Tumor-Associated Macrophages (TAMs) in HCC (GSE149614)

To investigate the immune landscape of HCC, we analyzed the discovery cohort dataset GSE149614 [22], which included paired normal and tumor tissues (*n* = 10). Unsupervised clustering of scRNA-seq profiles identified 36 distinct cellular subsets encompassing immune, stromal, and hepatocyte populations (Figure 1A). Whereas cytotoxic T cells, NK cells, and naïve B cells predominated in normal tissues, tumor samples displayed pronounced enrichment of TAMs, hepatocytes with metabolic alterations, and activated monocytes (Figure 1B). Quantification confirmed that TAMs accounted for nearly all of their cluster in tumor tissues, compared with only a small proportion in normal controls (Figure 1C). Among these, cluster 4 represented TAMs, characterized by high expression of SPP1 and canonical macrophage markers, such as CD68 and lysozyme (Figure 1D). Comparative analyses revealed a marked enrichment of *SPP1*^+^ TAMs in tumor tissues relative to normal liver (*p* < 0.0001). In the single-cell transcriptomic landscape, as shown in a dot plot profiling with the top 20 gene expression pattern (Figure 1E), cluster 4 emerged as a distinct *SPP1*^+^*BCL2A1*^+^ TAMs population. Bcl-2A1, a member protein of the Bcl-2 family, has been identified as pro-tumorigenic [25,26]. UMAP visualization (Figure 1F) revealed that this subset was uniquely enriched for the dual expression of *SPP1* and *BCL2A1*, a pattern largely absent from other clusters. Statistical analysis demonstrated significantly elevated *SPP1*^+^*BCL2A1*^+^ TAMs scores in tumor tissues (*p* < 0.0001) (Figure 1G), establishing this newly identified TAM subtype as an HCC-enriched immune subset.

### 3.2. Association of SPP1^+^BCL2A1^+^ TAMs Enrichment with Response to PD-1 Blockade (GSE151530)

To assess the clinical relevance of *SPP1*^+^*BCL2A1*^+^ TAM enrichment in the context of immunotherapy, we analyzed scRNA-seq profiles (Figure 2A) using the cohort dataset GSE151530 (*n* = 7) [21], which included patients treated with PD-1 blockade. Pre- and post-treatment scRNA-seq analyses revealed differential dynamics of immune cells (Figure 2B). Among these, dissimilar from clusters 8 and 18 with a monocyte/macrophage genotype, selective cluster 4 distinctly represented a TAM-like genotype and was identified in HCC before ICB treatment (Pre- versus Post-treatment) (Figure 2C). In search of *SPP1*^+^*BCL2A1*^+^ TAM expression, UMAP clustering analysis confirmed its expression was especially enriched in cluster 4 (Figure 2D). Before and after ICB treatment, intratumoral expression of *SPP1*^+^*BCL2A1*^+^ TAMs were not altered (Figure 2E). Notably, boxplots of *SPP1*^+^*BCL2A1*^+^ TAMs signature scores revealed significantly higher levels in non-responders (NRs) compared with responders (Rs) both before and after ICB treatment (Figure 2F). In the pre-treatment cohort, NR and R exhibited 21 and 38 positive cells, respectively (FDR-corrected *p* < 0.0001), whereas in the post-treatment cohort, the corresponding numbers were 20 and 7 (FDR-corrected *p* < 0.0013). To assess the robustness of these findings, a bootstrap-based power analysis was performed with 1000 iterations and a significance threshold of α = 0.05. The results demonstrated high statistical power, with estimated values of 1.000 for the pre-treatment and 0.966 for the post-treatment comparisons, confirming the reliability of the observed differences between NR and R groups. As considered by the limited cases, the exploratory observation data indicate that the persistence of *SPP1*^+^*BCL2A1*^+^ TAMs enrichment is associated with therapeutic resistance to PD-1 blockade.

### 3.3. Validation of SPP1^+^BCL2A1^+^ TAMs Enrichment and Its Significance Across Independent HCC Cohort in the ICB Setting (GSE206325)

Further validation of single-cell transcriptomes (Figure 3A) was performed using dataset GSE206325, which included paired normal and tumor tissues (*n* = 18) from HCC patients, post-treatment with ICB [23]. Consistent with the discovery dataset, UMAP clustering (Figure 3B) and proportional analyses (Figure 3C) revealed a substantial increase in the number of TAMs, characterized by cluster 20, in HCC tumor samples compared to the normal. Comparative boxplot analyses confirmed significantly elevated *SPP1*^+^*BCL2A1*^+^ TAM scores in tumor versus normal tissues (*p* < 0.05) (Figure 3D). These results establish that *SPP1*^+^*BCL2A1*^+^ TAM enrichment is a recurrent feature of HCC across multiple patient cohorts, supporting their role as a tumor-associated immune signature. Following ICB treatment, an increase in intratumoral *SPP1*^+^*BCL2A1*^+^ TAMs was identified in non-responders (NRs; *n* = 12) compared with responders (Rs; *n* = 6). Boxplots of *SPP1*^+^*BCL2A1*^+^ TAMs scores demonstrated significantly higher levels in non-responders (*p* < 0.01) (Figure 3E). The data confirm that the enrichment of *SPP1*^+^*BCL2A1*^+^ TAMs in tumors shows resistance to PD-1 blockade.

### 3.4. Distinct Enrichment of SPP1^+^BCL2A1^+^ TAMs Versus Cytotoxic T-Cell Responses in Non-Responders and Responders

We next examined the interplay between *SPP1*^+^*BCL2A1*^+^ TAMs, and other cell populations, as well as their potential link to the differences in ICB response. By using GSE206325, this analysis compares immune cell-type counts (Figure 4A) between non-responders (NRs; *n* = 12) and responders (Rs; *n* = 6). UMAP clustering (Figure 4B) and proportional analyses (Figure 4C) revealed a substantial increase in the number of TAMs, characterized by cluster 5, in non-responders (NRs) compared to responders (Rs). A striking divergence in the immune landscape (Figure 4D) revealed that *SPP1*^+^*BCL2A1*^+^-containing cluster 5 was significantly enriched in non-responders (NRs), accompanied by increased Treg subsets and reduced cytotoxic T-cell subsets. By contrast, responders (Rs) exhibited robust cytotoxic T-cell responses, particularly activated CD8^+^ effector and IFNG^+^ subsets, along with reduced expression of *SPP1*^+^*BCL2A1*^+^ TAMs, reflecting a favorable immune milieu. To explore the functional implications of *SPP1*^+^*BCL2A1*^+^ TAMs, we interrogated their transcriptional programs. KEGG pathway analyses demonstrated that enrichment for the cell survival-associated longevity pathway score was significantly higher in non-responders (NRs) compared to responders (Rs) to ICB therapy (*p* < 0.0001) (Figure 4E). These results suggest that the presence of specialized long-lived *SPP1*^+^*BCL2A1*^+^ TAMs is associated with resistance to ICB treatment, whereas lower pathway activity is linked to therapeutic response.

## 4. Discussion

*SPP1*^+^ TAMs are enriched in tumor tissues compared with normal liver [15,17,18,19,20,21], and their increased abundance is linked to worse overall survival as well as resistance to ICB therapy in HCC [16]. In this study, our integrative analyses across multiple scRNA-seq transcriptomic datasets further reveal that *SPP1*^+^*BCL2A1*^+^ TAMs are a distinct and clinically relevant immune subset in HCC. While *SPP1*^+^ TAMs are known to promote tumor progression through immunosuppressive mechanisms [15,16,17,18,19], the *SPP1*^+^*BCL2A1*^+^ TAM subset further exhibits a distinct transcriptional program associated with resistance to cell death. Elevated SPP1 expression reflects an anti-inflammatory, tissue-remodeling macrophage phenotype, whereas BCL2A1 confers a robust anti-apoptotic signature [26], collectively suggesting enhanced cellular survival potential. Our findings indicate that the newly identified *SPP1*^+^*BCL2A1*^+^ TAMs subtype represents a specialized macrophage displaying longevity-related pathways with potential roles in sustaining chronic inflammation, promoting matrix remodeling, and creating a survival-advantaged niche within the microenvironment in HCC, which may display resistance to ICB therapy.

The identification of *SPP1*^+^ TAMs in cancers as a prognostic biomarker has important clinical implications [13]. However, its subtypes, which share diverse pro-tumorigenic activities, remain undocumented. According to our findings, *SPP1*^+^*BCL2A1*^+^ TAMs were consistently enriched in tumor tissues compared with adjacent normal liver, and their abundance strongly correlated with poor response to PD-1 blockade. The application of current predictive markers for ICB in HCC and other cancers, including PD-L1 expression and tumor mutational burden [25], depends on the tumor type and the individual’s immune status. Our findings indicate that quantifying *SPP1*^+^*BCL2A1*^+^ TAM expression may provide a promising but preliminary candidate biomarker. Following a comprehensive validation, this could aid in selecting patients who are more likely to benefit from PD-1 blockade and in identifying those who may require combination strategies to overcome resistance.

Therapeutically, the presence of *SPP1*^+^*BCL2A1*^+^ TAMs raises the possibility of targeting this axis to enhance the efficacy of immunotherapy. In addition to direct blockade of SPP1 and BCL-2A1 by using depletion approach [16] and selective inhibitor targeting Bcl-2 family proteins [25,27], several approaches, including (1) blocking cell recruitment, (2) depleting cells within the TME, (3) enhancing their phagocytic activity, (4) reprogramming cell functions, (5) addressing its heterogeneity, (6) modulating cell metabolism, and (7) employing genetically engineered macrophages, are under investigation and could be adapted to target *SPP1^+^BCL2A1^+^* TAMs specifically [28,29]. In addition to sustaining cellular survival, the Bcl-2 inhibitor APG-2575 enhances anti-PD-1 efficacy by reprogramming M2-like macrophages into an M1 phenotype, thereby boosting CCL5/CXCL10 secretion, restoring T-cell function, and improving immunotherapy response in mouse models [30]. However, BCL2A1 has structural resistance to BH3-mimetics and may require indirect targeting strategies. Further preclinical studies of SPP1 and BCL2A1 targeting will be required to test these strategies in HCC models using ICB combinational therapy.

In macrophages, SPP1 drives pro-tumorigenic functions by promoting angiogenesis, extracellular matrix remodeling, and immunosuppression [13,14,15,16]. Our study demonstrates that these features extend to the immunotherapy setting, where the presence of *SPP1*^+^*BCL2A1*^+^ TAMs is highly associated with increased Treg responses and impaired T-cell responses, particularly in non-responders to ICB therapy in HCC patients. These findings are partly consistent with prior work implicating *SPP1*^+^ TAMs in tumor progression and resistance to ICB therapy in HCC [16]. Mechanistically, *SPP1*^+^ TAMs interact with cancer-associated fibroblasts (CAFs) at the invasive margins via SPP1–integrin/CD44 signaling, driving CAF activation, ECM deposition, and chemokine release, which establishes a stromal barrier that excludes CD8^+^ T cells and promotes ICB resistance. Currently, TREM2 (triggering receptor expressed on myeloid cells 2)^+^ TAMs drive tumor progression and immunotherapy resistance in HCC through SPP1-mediated effects on both cancer cells and CD8^+^ T cells [31]. This suggests the potential involvement of *SPP1*^+^*BCL2A1*^+^ TAMs in the development of resistance to ICB immunotherapy.

## 5. Limitations

Our study has limitations. While the analyses incorporated multiple independent cohorts, the sample sizes of immunotherapy datasets remain modest, limiting generalizability. Moreover, scRNA-seq datasets capture transcriptional states at a single time point and may not fully reflect the dynamic changes that occur during therapy. Future studies should incorporate longitudinal sampling and spatial transcriptomics to better define the temporal and spatial interactions between *SPP1*^+^*BCL2A1*^+^ TAMs, and T-cell subsets. Additionally, performing immunohistochemistry and multiplex immunofluorescence staining on an independent HCC tissue cohort is speculated to confirm the co-expression of SPP1 and BCL2A1 within CD68^+^/CD163^+^ macrophages and to assess their spatial association with exhausted T cells. Finally, functional validation in experimental models will be necessary to establish causal relationships between *SPP1*^+^*BCL2A1*^+^ TAMs and ICB resistance. As a link between longevity-related pathways and macrophage survival as correlative rather than causal, emphasizing that mechanistic experiments are required to confirm whether BCL2A1 directly promotes TAM persistence.

## 6. Conclusions

We identify intratumoral *SPP1*^+^*BCL2A1*^+^ TAMs as a promising but preliminary candidate biomarker of poor prognosis in HCC patients receiving PD-1 blockade. The presence of *SPP1*^+^*BCL2A1*^+^ TAMs is associated with an immunosuppressive tumor microenvironment in non-responders. *SPP1*^+^*BCL2A1*^+^ TAMs emerge as both predictive markers of therapeutic response and potential therapeutic targets, warranting future validation in larger prospective cohorts and functional studies exploring macrophage–T-cell interactions. These findings advance our understanding of the HCC immune microenvironment and open new avenues for improving immunotherapy outcomes in this challenging disease.

## Figures and Tables

**Figure 1 diagnostics-15-02680-f001:**
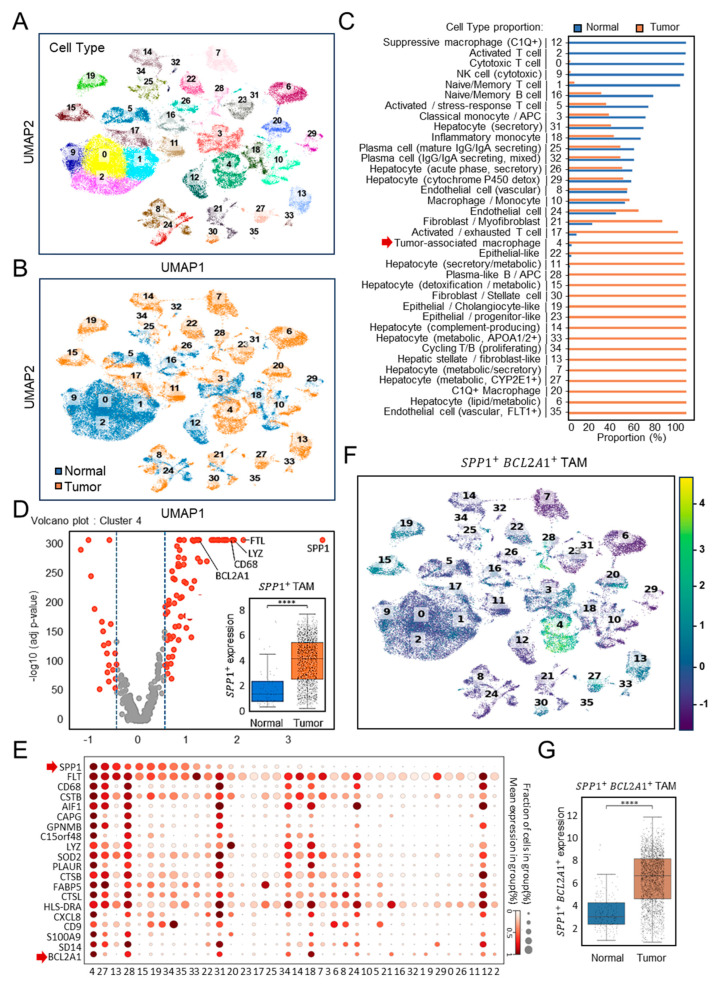
Single-cell transcriptomic profiling of *SPP1*^+^*BCL2A1*^+^ TAMs in the HCC discovery cohort (GSE149614). (**A**) UMAP plots showing the clustering of immune and non-immune cell subsets. (**B**) UMAP plots of paired (*n* = 10) normal versus tumor tissues. Cell identities were annotated using canonical marker genes. (**C**) Quantification of cell-type proportions between tumor and normal samples. Cluster 4 represented TAMs (red arrow). (**D**) Volcano plot showing high-dimensional gene expression in cluster 4 of *SPP1*^+^ TAMs. Boxplot shows *SPP1*^+^ TAM scores comparing normal versus tumor samples. **** *p* < 0.0001. (**E**) scRNA-seq dot plot illustrating the expression of the top 20 canonical marker genes across immune cell clusters, highlighting *SPP1*^+^ and *BCL2A1*^+^ gene expression (red arrow) mainly enriched in cluster 4. (**F**) UMAP plots showing the clustering of *SPP1*^+^*BCL2A1*^+^ TAMs, specifically enriched in cluster 4, in paired normal versus tumor tissues. (**G**) Quantification of *SPP1*^+^*BCL2A1*^+^ TAMs between tumor and normal samples. **** *p* < 0.0001.

**Figure 2 diagnostics-15-02680-f002:**
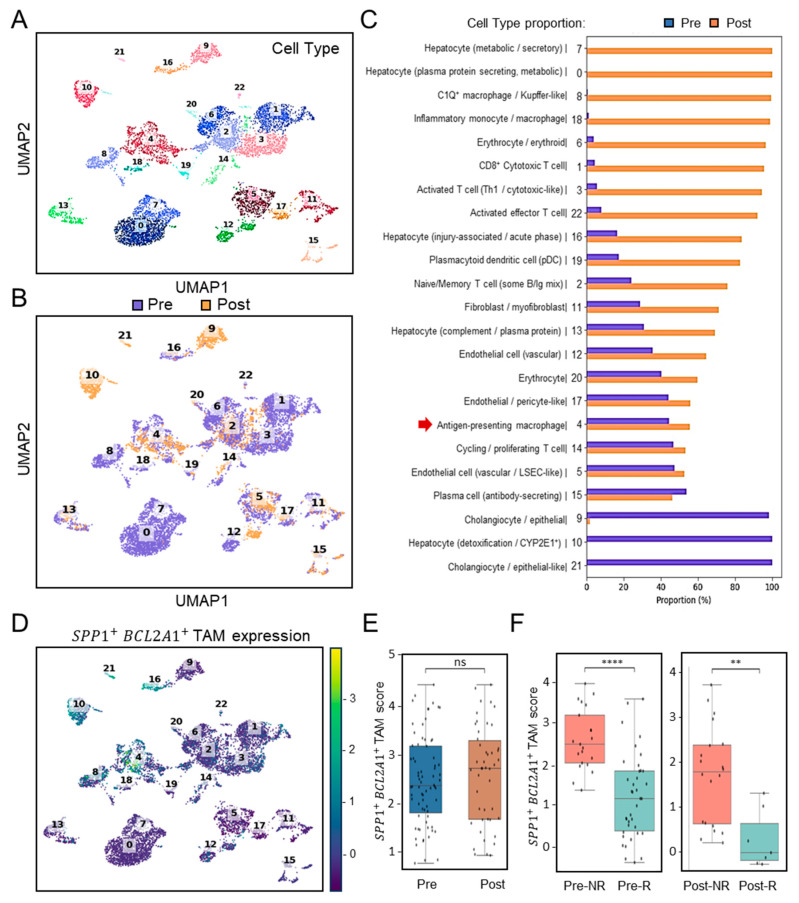
Single-cell transcriptomic profiling and association of *SPP1*^+^*BCL2A1*^+^ TAMs enrichment in HCC patients with ICB response (GSE151530). (**A**) UMAP plot showing clustering of immune and non-immune cell subsets from paired (*n* = 7) pre- and post-treatment samples. (**B**) UMAP plot highlighting pre- (blue) and post-treatment (orange) cell distribution. (**C**) Quantification of cell-type proportions between pre- and post-treatment samples. Notably, antigen-presenting macrophages (cluster 4, red arrow) were enriched post-treatment. (**D**) UMAP plot displaying *SPP1*^+^*BCL2A1*^+^ TAM expression, mainly identified in cluster 4, across pre- and post-treatment tissues. (**E**) Boxplot quantifying *SPP1*^+^*BCL2A1*^+^ TAMs scores between pre- and post-treatment groups. ns, not significant. (**F**) Stratified analysis of *SPP1*^+^*BCL2A1*^+^ TAMs scores in non-responders (NRs; *n* = 1; Pre versus Post, 21 and 20 positive cells) and responders (Rs; *n* = 3; Pre versus Post, 38 and 7 positive cells) pre- and post-treatment. **** *p* < 0.0001, ** *p* < 0.01.

**Figure 3 diagnostics-15-02680-f003:**
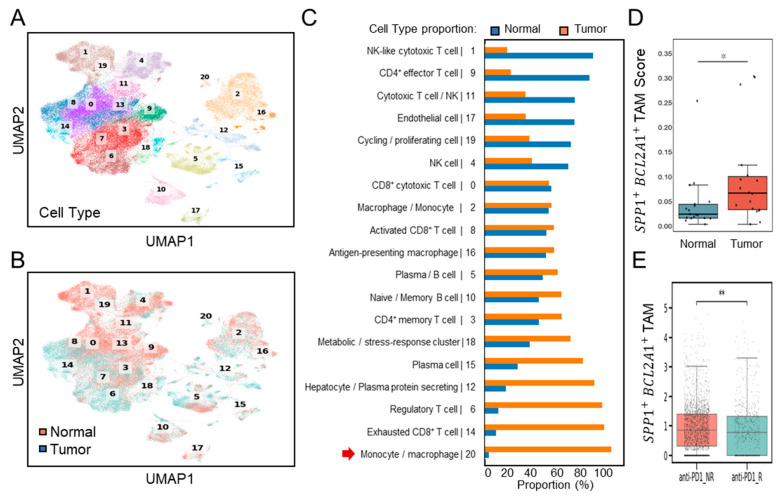
Single-cell transcriptomic profiling of *SPP1*^+^*BCL2A1*^+^ TAMs in an independent HCC cohort (GSE206325) for validation. (**A**) UMAP plot showing clustering of immune and non-immune cell subsets. (**B**) UMAP plot highlighting paired (*n* = 18) normal and tumor cell distribution. (**C**) Quantification of cell-type proportions between normal and tumor samples, with enrichment of monocyte/macrophage cluster 20 (red arrow) in tumor tissues. (**D**) Boxplot showing *SPP1*^+^*BCL2A1*^+^ TAMs scores in normal versus tumor tissues. * *p* < 0.05. (**E**) Boxplot showing *SPP1*^+^*BCL2A1*^+^ TAM levels in anti–PD-L1 non-responders (NRs; *n* = 12) and responders (Rs; *n* = 6). ** *p* < 0.01.

**Figure 4 diagnostics-15-02680-f004:**
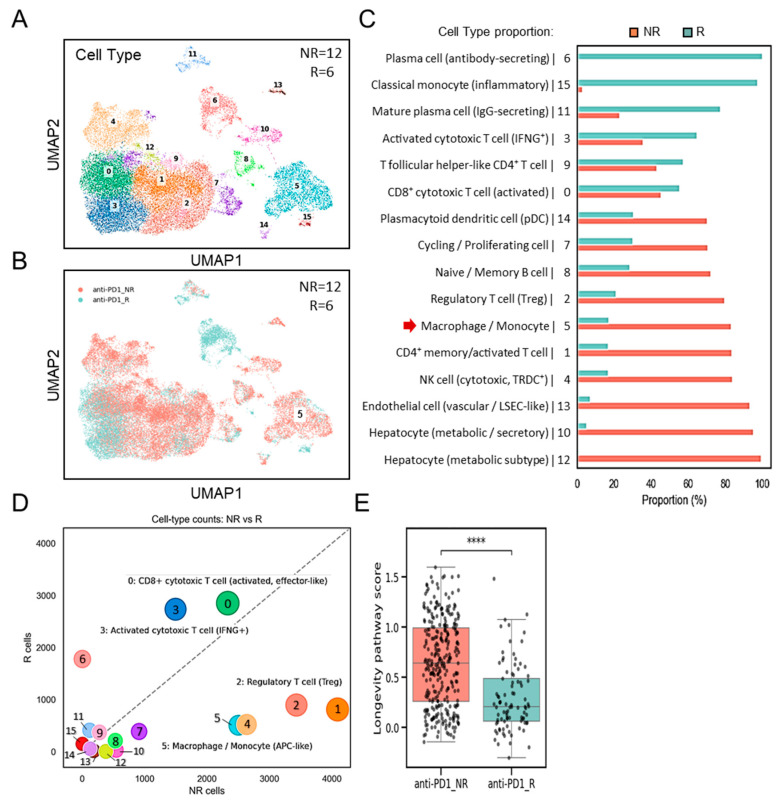
Single-cell transcriptomic analysis of immune subsets and their possible regulation in non-responders (NRs) versus responders (Rs) to anti–PD-1 therapy (GSE206325). (**A**) UMAP plot showing clustering of immune cell subsets from non-responders (NRs; *n* = 12) and responders (Rs; *n* = 6). (**B**) UMAP plot highlighting the distribution of cells from the NR and R groups. (**C**) Quantification of cell-type proportions between NR and R samples, with enrichment of macrophage/monocyte cluster 5 (red arrow) in the NR group. (**D**) Association analysis comparing cell-type counts between NR and R, showing enrichment of CD8^+^ cytotoxic T cells (activated, effector-like and IFNG^+^ subsets), regulatory T cells (Tregs), and macrophage/monocyte (containing *SPP1*^+^*BCL2A1*^+^ TAMs) subsets. (**E**) Boxplot showing longevity pathway scores in anti–PD-1 NR versus R groups. **** *p* < 0.0001.

## Data Availability

The datasets generated during and/or analyzed during the current study are available from the corresponding author on reasonable request. All datasets analyzed in this study are publicly accessible through the National Center for Biotechnology Information Gene Expression Omnibus (GEO), including GSE149614, GSE151530 and GSE206325.

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
