# Peer review of "Intratumoral SPP1+BCL2A1+ Tumor-Associated Macrophages Predict Poor Response to PD1 Blockade"

_diagnostics, 2025, doi:10.3390/diagnostics15212680_

Round 1
Reviewer 1 Report
Comments and Suggestions for Authors
The manuscript presents an integrative single-cell RNA-seq analysis across three public hepatocellular carcinoma (HCC) cohorts (GSE149614, GSE151530, GSE206325) to identify a novel intratumoral macrophage subset co-expressing SPP1 and BCL2A1. They show that this SPP1+BCL2A1+ TAM population is enriched in tumor tissues and significantly associated with resistance to PD-1 blockade, suggesting both a prognostic biomarker and potential therapeutic target.
The manuscript is well written. Use of multiple independent scRNA-seq databases increases robustness. Integration of gene signature scoring, UMAP visualization and KEGG pathway analysis offers mechanistic insights.
Major comments:
- The manuscript relies entirely on transcriptomic datasets without functional validation. Validation by immunohistochemistry or flow cytometry to confirm co-expression of SPP1 and BCL2A1 in TAMs—and demonstration of their immunosuppressive function—would substantially strengthen the conclusions.
- The GSE151530 dataset contains only four patients (one non-responder vs. three responders), limiting statistical power. The authors should more thoroughly discuss how such small numbers affect reliability and consider aggregating additional ICB-treated HCC datasets or carrying out power calculations.
- Clustering criteria are not clear. The authors mention using canonical markers and ChatGPT for cell-type identification, the criteria for cluster assignment require clarification. Please specify which lineage markers defined each major immune subset and provide a supplementary table with marker genes per cluster.
- Statistical testing. The methods describe Wilcoxon tests and FDR thresholds, but it is unclear how comparisons of signature scores between responders and non-responders account for multiple testing. Clarify whether p-values in Figures 2F, 3E, and 4E are adjusted and describe any corrections applied
- The link between longevity pathway scores and TAM survival is correlative. The Discussion should temper causal language and indicate that functional studies are needed to confirm whether BCL2A1 directly confers prolonged TAM viability
Minor comments:
- Consider rephrasing the title to " Intratumoral SPP1⁺BCL2A1⁺ Tumor-Associated Macrophages Predict Poor Response to PD-1 Blockade" for clarity
- State whether patient metadata (age, etiology, prior treatments) were available and, if so, whether they were balanced between responders and non-responders.
Author Response
Please refer to the attached REPLY LETTER.

Reviewer 2 Report
Comments and Suggestions for Authors
This is a well-written and timely study addressing the urgent need for better biomarkers in HCC immunotherapy. I was particularly impressed by your use of multiple independent scRNA-seq cohorts to identify and validate the SPP1+BCL2A1+ TAM signature. However, there are suggestions for improvement:
- Please clarify the statements on whether a high SPP1 pattern is linked to prolonged survival or resistance to immunotherapy.
- Consider strengthening the rationale by explaining the specific novelty of investigating BCL2A1 in addition to the already-known SPP1+ TAM signature.
- The transition from the general role of diverse TAMs to the specific focus on SPP1-expressing macrophages could be smoother to improve narrative flow
- The stated use of ChatGPT for marker interpretation is not a reproducible or scientifically validated method. This foundational step must be based exclusively on citable, peer-reviewed literature and established databases to ensure the validity of all subsequent findings
- The manuscript states that gene signature scoring was used to quantify the enrichment of the TAM population of interest, however, the methods section fails to provide the exact list of genes that constitute this signature. For the results to be reproducible, the complete gene list used must be provided
- The section on pathway analysis mentions a "particular focus... on the longevity-regulating pathway" but does not define it. Please elaborate
- The statistical analysis section lists the Mann-Whitney U test as a method for comparison. This test is not statistically valid or meaningful for the responder (R; n=3) vs. non-responder (NR; n=1) comparison in the GSE151530 cohort. The methods section should acknowledge this limitation and clarify that this specific comparison can only be descriptive, not statistically significant
- I suggest that all claims of statistical significance in this section and in Figure 2F must be removed. The section should be rewritten to present the findings as a preliminary, descriptive observation from a small case series that requires validation
- The discussion proposes that quantifying SPP1+BCL2A1+ TAMs could provide a "more reliable means of stratifying patients". Given the small sample sizes and the correlational nature of the study, this language is too strong. The findings are promising but preliminary. Please readjust the tone.
- The manuscript describes the SPP1+BCL2A1+ TAMs as having a signature "consistent with inflammatory macrophages". This term is ambiguous. SPP1+ TAMs are often associated with pro-tumor, M2-like, and immunosuppressive functions rather than a classic pro-inflammatory, anti-tumor state. Please use more precise terminology to define the likely functional polarization of these cells.
- The discussion exclusively cites literature that supports the role of SPP1+ TAMs in promoting immunotherapy resistance. However, the introduction itself cites a study suggesting that high SPP1 is associated with prolonged survival following anti-PD-L1 therapy. A robust scientific discussion must acknowledge and attempt to reconcile such conflicting findings in the field.
- The discussion suggests targeting the BCL2A1 protein as a potential therapeutic strategy. However, it fails to mention that BCL2A1 is known to be a particularly difficult member of the Bcl-2 family to target with existing small-molecule inhibitors. Please acknowledge this therapeutic challenge rigorously.
- A strong narration should contextualize the findings by addressing the study's weaknesses. This section largely overlooks the most significant limitations, primarily the statistically invalid analysis of the GSE151530 cohort.
- The conclusion describes the SPP1+BCL2A1+ TAM signature as a "robust biomarker". This claim is too strong. The term should be changed to "promising candidate biomarker" or "potential biomarker".
- The text states, "By shaping an immunosuppressive tumor microenvironment, SPP1+BCL2A1+ TAMs emerge as...". This phrasing implies a proven causal mechanism. The study only demonstrates an association between the presence of these TAMs and an immunosuppressive environment in non-responders. The language should be modified to reflect this correlation (e.g., "The presence of these TAMs is associated with...")
- The conclusion presents the findings as a final statement without acknowledging the necessary next steps. A strong conclusion should briefly mention the need for future research, such as validation in larger, prospective patient cohorts and functional studies to confirm the mechanistic role of these cells in ICB resistance
Author Response

(The authors gave the same response as above.)

Round 2
Reviewer 1 Report
Comments and Suggestions for Authors
The manuscript is suitable for publication